# Which Mutual-Information Representation Learning Objectives are Sufficient for Control?

**Kate Rakelly**[*]    **Abhishek Gupta**    **Carlos Florensa**    **Sergey Levine**

University of California, Berkeley

{rakelly, abhigupta, florensacc, svlevine}@eecs.berkeley.edu

## Abstract

Mutual information (MI) maximization provides an appealing formalism for learning representations of data. In the context of reinforcement learning (RL), such representations can accelerate learning by discarding irrelevant and redundant information, while retaining the information necessary for control. Much prior work on these methods has addressed the practical difficulties of estimating MI from samples of high-dimensional observations, while comparatively less is understood about *which* MI objectives yield representations that are sufficient for RL from a theoretical perspective. In this paper, we formalize the sufficiency of a state representation for learning and representing the optimal policy, and study several popular MI based objectives through this lens. Surprisingly, we find that two of these objectives can yield insufficient representations given mild and common assumptions on the structure of the MDP. We corroborate our theoretical results with empirical experiments on a simulated game environment with visual observations.

## 1   Introduction

Deep reinforcement learning (RL) algorithms are in principle capable of learning policies from high-dimensional observations, such as camera images [49, 39, 32]. However, policy learning in practice faces a bottleneck in acquiring useful representations of the observation space [58]. State representation learning approaches aim to remedy this issue by learning structured and compact representations on which to perform RL. A useful state representation should be *sufficient* to learn and represent the optimal policy or the optimal value function, while discarding irrelevant and redundant information. Understanding whether or not an objective is guaranteed to yield sufficient representations is important, because insufficient representations make it impossible to solve certain problems. For example, an autonomous vehicle would not be able to navigate safely if its state representation did not contain information about the color of the stoplight in front of it. With the increasing interest in leveraging offline datasets to learn representations for RL [19, 35, 64], the question of sufficiency becomes even more important to understand if the representation is capable of representing policies and value functions for downstream tasks.

While a wide range of representation learning objectives have been proposed in the literature [41], in this paper we focus on analyzing representations learned by maximizing the mutual information (MI) between random variables. Prior work has proposed many different MI objectives involving the variables of states, actions, and rewards at different time-steps [4, 52, 53, 58]. While much prior work has focused on how to optimize these various MI objectives in high dimensions [61, 8, 52, 27], we focus instead on their ability to yield theoretically sufficient representations. We find that two commonly used objectives are insufficient for the general class of MDPs, in the most general case, and prove that another typical objective is sufficient. We illustrate the analysis with both didactic examples in which MI can be computed exactly and deep RL experiments in which we approximately maximize

---

[*]Correspondence to rakelly@eecs.berkeley.edu

35th Conference on Neural Information Processing Systems (NeurIPS 2021).

the MI objective to learn representations of visual inputs. The experimental results corroborate our theoretical findings, and demonstrate that the sufficiency of a representation can have a substantial impact on the performance of an RL agent that uses that representation. This paper provides guidance to the deep RL practitioner on when and why objectives may work well or fail, and also provides a formal framework to analyze newly proposed representation learning objectives based on MI.

## 2 Related Work

In this paper, we analyze several widely used mutual information objectives for control. In this section we first review MI-based unsupervised learning, then the application of these techniques to the RL setting. Finally, we discuss alternative perspectives on representation learning in RL.

**Mutual information-based unsupervised learning.** A common technique for unsupervised representation learning based on the InfoMax principle [43, 9] is to maximize the MI between the input and its latent representation subject to domain-specific constraints [7]. This technique has been applied to learn representations for natural language [14], video [65], and images [5, 27] and even policy learning via RL in high dimensions [62]. To address the difficulties of estimating MI from samples [45] and with high-dimensional inputs [61], much recent work has focused on improving MI estimation via variational methods [61, 55, 52, 8]. In this work we are concerned with analyzing the MI objectives, and not the estimation method. In our experiments with image observations, we use noise contrastive estimation methods [24], though other choices could also suffice.

**Mutual information objectives in RL.** RL adds aspects of temporal structure and control to the standard unsupervised learning problem discussed above (see Figure 1). This structure can be leveraged by maximizing MI between sequential states, actions, or combinations thereof. Some works omit the action, maximizing the MI between current and future states [4, 52, 64]. Several prior works [51, 57, 60, 44] maximize MI objectives that closely resemble the $\mathbb{J}_{fwd}$ objective we introduce in Section 4, while others optimize related objectives by learning latent forward dynamics models [70, 33, 74, 26, 39]. Multi-step inverse models, closely related to the $\mathbb{J}_{inv}$ objective (Section 4), have been used to learn control-centric representations [71, 23]. Single-step inverse models have been deployed as regularization of forward models [73, 2] and as an auxiliary loss for policy gradient RL [58, 53]. Our result regarding the sufficiency of this objective is similar to an example explored in Misra et al. [48]; however, we relate ours to MI objectives. The MI objectives that we study have also been used as reward bonuses to improve exploration, without impacting the representation, in the form of empowerment [37, 36, 50, 40] and information-theoretic curiosity [63].

**Representation learning for reinforcement learning.** In RL, the problem of finding a compact state space has been studied as state aggregation or abstraction [6, 42]. Abstraction schemes include bisimulation [22], homomorphism [56], utile distinction [46], and policy irrelevance [30]. While efficient algorithms exist for MDPs with known transition models for some abstraction schemes such as bisimulation [18, 22], in general obtaining error-free abstractions is impractical for most problems of interest. For approximate abstractions prior work has bounded the sub-optimality of the policy [11, 13, 1] as well as the sample efficiency [38, 68, 16], with some results in the deep learning setting [21, 51]. In this paper, we focus on whether a representation can be used to learn the optimal policy, and not the tractability of learning. Li et al. [42] shares this focus; while they establish convergence properties of $Q$-learning with representations satisfying different notions of sufficiency, we leverage their $Q^*$-sufficiency criteria to evaluate representations learned via MI-based objectives. Alternative approaches to representation learning for RL include priors based on the structure of the physical world [31] or heuristics such as disentanglement [67], meta-learning general value functions [69], predicting multiple value functions [10, 17, 29] and predicting domain-specific measurements [47, 15]. We restrict our analysis to objectives that can be expressed as MI-maximization. In our paper we focus on the representation learning problem, disentangled from exploration, a strategy shared by prior works [19, 35, 64, 72].

## 3 Representation Learning for RL

The goal of representation learning for RL is to learn a compact representation of the state space that discards irrelevant and redundant information, while still retaining sufficient information to represent policies and value functions needed for learning. In this section we formalize this problem, and propose and define the concept of sufficiency to evaluate the usefulness of a representation.

### 3.1 Preliminaries

We begin with brief preliminaries of reinforcement learning and mutual information.

**Reinforcement learning.** A Markov decision process (MDP) is defined by the tuple $(\mathcal{S}, \mathcal{A}, \mathcal{T}, r)$, where $\mathcal{S}$ is the set of states, $\mathcal{A}$ the set of actions, $\mathcal{T} : \mathcal{S} \times \mathcal{A} \times \mathcal{S} \to [0, 1]$ the state transition distribution, and $r : \mathcal{S} \to \mathbb{R}$ the reward function [2]. We will use capital letters to refer to random variables and lower case letters to refer to values of those variables (e.g., $S$ is the random variable for the state and $\mathbf{s}$ is a specific state). Throughout our analysis we will often be interested in multiple reward functions, and denote a set of reward functions as $\mathcal{R}$. The objective of RL is to find a policy that maximizes the sum of discounted returns $\bar{R}$ for a given reward function $r$, and we denote this optimal policy as $\pi_r^* = \arg\max_\pi \mathbb{E}_\pi[\sum_t \gamma^t r(S_t, A_t)]$ for discount factor $\gamma$. We also define the optimal $Q$-function as $Q_r^*(\mathbf{s}_t, \mathbf{a}_t) = \mathbb{E}_{\pi^*}[\sum_{t=1}^\infty \gamma^t r(S_t, A_t)|\mathbf{s}_t, \mathbf{a}_t]$. The optimal $Q$-function satisfies the recursive Bellman equation, $Q_r^*(\mathbf{s}_t, \mathbf{a}_t) = r(\mathbf{s}_t, \mathbf{a}_t) + \gamma\mathbb{E}_{p(\mathbf{s}_{t+1}|\mathbf{s}_t, \mathbf{a}_t)} \arg\max_{\mathbf{a}_{t+1}} Q_r^*(\mathbf{s}_{t+1}, \mathbf{a}_{t+1})$. An optimal policy and the optimal Q-function are related according to $\pi^*(\mathbf{s}) = \arg\max_\mathbf{a} Q^*(\mathbf{s}, \mathbf{a})$.

**Mutual information.** In information theory, the mutual information (MI) between two random variables, $X$ and $Y$, is defined as [12]:

$$I(X; Y) = \mathbb{E}_{p(x,y)} \log \frac{p(x, y)}{p(x)p(y)} = H(X) - H(X|Y). \tag{1}$$

The first definition indicates that MI can be understood as a relative entropy (or KL-divergence), while the second underscores the intuitive notion that MI measures the reduction in the uncertainty of one random variable from observing the value of the other.

**Representation learning for RL.** While state aggregation methods typically define deterministic rules to group states in the representation [6, 42], MI-based representation learning methods used for deep RL treat the representation as a random variable [51, 52, 53]. Accordingly, we formalize a representation as a stochastic mapping between original state space and representation space.

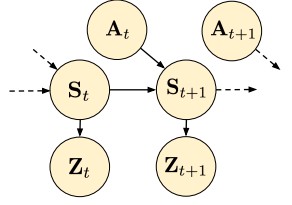

**Definition 1.** *A **stochastic representation** $\phi_\mathcal{Z}(\mathbf{s})$ is a mapping from states $\mathbf{s} \in \mathcal{S}$ to a probability distribution $p(Z|S = \mathbf{s})$ over elements of a new representation space $z \in \mathcal{Z}$.*

Figure 1: Probabilistic graphical model illustrating the state representation learning problem: estimate representation $Z$ from original state $S$.

In this work we consider learning state representations from data by maximizing an objective $\mathbb{J}$. Given an objective $\mathbb{J}$, we define the set of representations that maximize this objective as $\Phi_\mathbb{J} = \{\phi_\mathcal{Z} \mid \phi_\mathcal{Z} \in \arg\max \mathbb{J}(\phi)\}$. Unlike problem formulations for partially observed settings [70, 26, 39], we assume that $S$ is a Markovian state; therefore the representation for a given state is conditionally independent of the past states, a common assumption in the state aggregation literature [6, 42]. See Figure 1 for a depiction of the graphical model.

### 3.2 Sufficient Representations for Reinforcement Learning

We now turn to the problem of evaluating stochastic representations for RL. Intuitively, we expect a useful state representation to be capable of representing an optimal policy in the original state space.

**Definition 2.** *A representation $\bar{\phi}_\mathcal{Z}$ is $\pi^*$-**sufficient** with respect to a set of reward functions $\mathcal{R}$ if $\forall r \in \mathcal{R}$, $\phi_\mathcal{Z}(\mathbf{s}_1) = \phi_\mathcal{Z}(\mathbf{s}_2) \implies \pi_r^*(A|\mathbf{s}_1) = \pi_r^*(A|\mathbf{s}_2)$.*

When a stochastic representation $\phi_\mathcal{Z}$ produces the same distribution over the representation space for two different states $\mathbf{s}_1$ and $\mathbf{s}_2$ we say it *aliases* these states. Unfortunately, as already proven in Theorem 4 of Li et al. [42] for the more restrictive case of deterministic representations, being able to represent the optimal policy does not guarantee that it can be learned via RL in the representation space. Accordingly, we define a stricter notion of sufficiency that *does* guarantee the convergence of

---

[2]We restrict our attention to MDPs where the reward can be expressed as a function of the state, which is fairly standard across a broad set of real world RL problems

Q-learning to the optimal policy in the original state space (refer to Theorem 4 of Li et al. [42] for the proof of this).

**Definition 3.** *A representation $\phi_{\mathcal{Z}}$ is $Q^*$-**sufficient** with respect to a set of reward functions $\mathcal{R}$ if $\forall r \in \mathcal{R}$, $\phi_{\mathcal{Z}}(\mathbf{s}_1) = \phi_{\mathcal{Z}}(\mathbf{s}_2) \implies \forall \mathbf{a}, Q_r^*(\mathbf{a}, \mathbf{s}_1) = Q_r^*(\mathbf{a}, \mathbf{s}_2)$.*

Note that $Q^*$-sufficiency implies $\pi^*$-sufficiency since an optimal policy can be recovered from the optimal Q-function via $\pi_r^*(s) = \arg\max_a Q_r^*(s, a)$ [66]; however the converse is not true. We emphasize that while $Q^*$-sufficiency guarantees convergence, it does not guarantee tractability, which has been explored in prior work [38, 16].

We will further say that an *objective* $\mathbb{J}$ is sufficient with respect to some set of reward functions $\mathcal{R}$ if all the representations that maximize that objective $\Phi_{\mathbb{J}}$ are sufficient with respect to every element of $\mathcal{R}$ according to the definition above. Surprisingly, we will demonstrate that not all commonly used objectives satisfy this basic qualification.

## 4 Mutual Information for Representation Learning in RL

In our study, we consider several MI objectives proposed in the literature.

**Forward information:** A commonly sought characteristic of a state representation is to ensure it retains maximum predictive power over future state representations. This property is satisfied by representations maximizing the following MI objective,

$$\mathbb{J}_{fwd} = I(Z_{t+1}; Z_t, A_t) = H(Z_{t+1}) - H(Z_{t+1}|Z_t, A_t). \tag{2}$$

We suggestively name this objective "forward information" due to the second term, which is the entropy of the forward dynamics distribution. This objective and closely related ones have been used in prior works [51, 57, 60, 44].

**State-only transition information:** Several popular methods [52, 4, 64] optimize a similar objective, but do not include the action:

$$\mathbb{J}_{state} = I(Z_{t+k}; Z_t) = H(Z_{t+k}) - H(Z_{t+k}|Z_t). \tag{3}$$

As we will show, the exclusion of the action can have a profound effect on the characteristics of the resulting representations.

**Inverse information:** Another commonly sought characteristic of state representations is to retain maximum predictive power of the action distribution that could have generated an observed transition from $\mathbf{s}_t$ to $\mathbf{s}_{t+1}$. Such representations can be learned by maximizing the following information theoretic objective:

$$\mathbb{J}_{inv} = I(A_t; Z_{t+k}|Z_t) = H(A_t|Z_t) - H(A_t|Z_t, Z_{t+k}) \tag{4}$$

We suggestively name this objective "inverse information" due to the second term, which is the entropy of the inverse dynamics. A wide range of prior work learns representations by optimizing closely related objectives [23, 58, 2, 53, 71, 73]. Intuitively, inverse models allow the representation to capture only the elements of the state that are necessary to predict the action, allowing the discard of potentially irrelevant information.

## 5 Sufficiency Analysis

In this section we analyze the sufficiency of representations obtained by maximizing each objective presented in Section 4. To focus on the representation learning problem, we decouple it from RL by assuming access to a dataset of transitions collected with a policy that reaches all states with non-zero probability, which can then be used to learn the desired representation. We also assume that distributions, such as the dynamics or inverse dynamics, can be modeled with arbitrary accuracy, and that the maximizing set of representations for a given objective can be computed. While these assumptions might be relaxed in any practical RL algorithm, and exploration plays a confounding role, the ideal assumptions underlying our analysis provide the best-case scenario for objectives to yield provably sufficient representations. In other words, objectives found to be provably insufficient under ideal conditions will continue to be insufficient under more realistic assumptions.

## 5.1 Forward Information

In this section we show that a representation that maximizes $\mathbb{J}_{fwd}$ is sufficient for optimal control under any reward function. This result aligns with the intuition that a representation that captures forward dynamics can represent everything predictable in the state space, and can thus be used to learn the optimal policy for any task. This strength can also be a weakness if there are many predictable elements that are irrelevant for downstream tasks, since the representation retains more information than is needed for the task. Note that the representation can still discard information in the original state, such as independent random noise at each timestep.

**Proposition 1.** $\mathbb{J}_{fwd}$ *is sufficient for all reward functions.*

*Proof. (Sketch)* We first show in Lemma 1 that if $Z_t, A_t$ are maximally informative of $Z_{t+1}$, they are also maximally informative of the return $\bar{R}_t$. Thanks to the Markov structure, we then show in Lemma 2 that $\mathbb{E}_{p(Z_t|S_t=\mathbf{s})} p(\bar{R}_t|Z_t, A_t) = p(\bar{R}_t|S_t = \mathbf{s}, A_t)$. In other words, given $\phi_{\mathcal{Z}}$, additionally knowing $S$ doesn't change our belief about the future return. The $Q$-value is the expectation of the return, so $Z$ has as much information about the $Q$-value as $S$. See Appendix A.1 for the proof. □

## 5.2 State-Only Transition Information

While $\mathbb{J}_{state}$ is closely related to $\mathbb{J}_{fwd}$, we now show that $\mathbb{J}_{state}$ is not sufficient.

**Proposition 2.** $\mathbb{J}_{state}$ *is not sufficient for all reward functions.*

*Proof.* We show this by counter-example with the deterministic-transition MDP defined in Figure 2 (left). For all four states, let the two actions $\mathbf{a}_0$ and $\mathbf{a}_1$ be equally likely under the policy distribution. In this case, each state gives no information about which of the two possible next states is more likely; this depends on the action. Therefore, a representation maximizing $\mathbb{J}_{state}$ is free to alias states with the same next-state distribution, such as $\mathbf{s}_0$ and $\mathbf{s}_3$. An alternative view is that such a representation can maximize $\mathbb{J}_{state} = H(Z_{t+k}) - H(Z_{t+k}|Z_t)$ by reducing both terms in equal amounts - aliasing $\mathbf{s}_0$ and $\mathbf{s}_3$ decreases the marginal entropy as well as the entropy of predicting the next state starting from $\mathbf{s}_1$ or $\mathbf{s}_2$. However, this aliased representation is not capable of representing the optimal policy, which must distinguish $\mathbf{s}_0$ and $\mathbf{s}_3$ in order to choose the correct action to reach $\mathbf{s}_2$, which yields reward. □

In Figure 2 (right), we illustrate the insufficiency of $\mathbb{J}_{state}$ computationally, by computing the values of $\mathbb{J}_{state}$ and $\mathbb{J}_{fwd}$ for different state representations of the above MDP, ordered by decreasing compression (increasing $I(Z; S)$) left to right. At the far left of the plot is the representation that

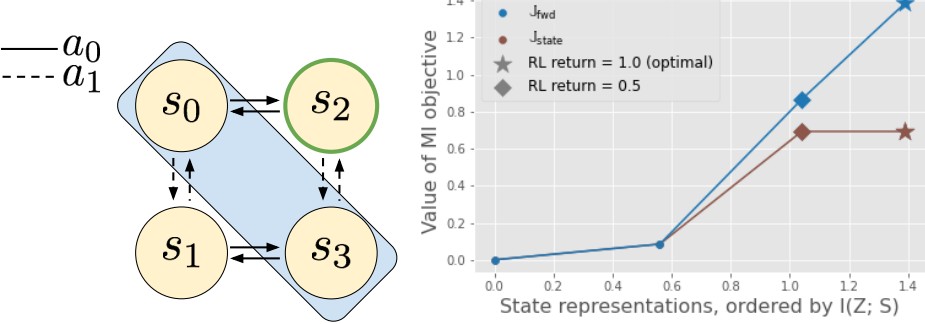

Figure 2: (left) A representation that aliases the states $\mathbf{s}_0$ and $\mathbf{s}_3$ into a single state maximizes $\mathbb{J}_{state}$ but is not sufficient to represent the optimal policy which must choose different actions in $\mathbf{s}_0$ and $\mathbf{s}_3$ to reach $\mathbf{s}_2$ which yields reward. (right) Values of $\mathbb{J}_{state}$ and $\mathbb{J}_{fwd}$ for a few representative state representations, ordered by increasing $I(Z; S)$. The representation that aliases $\mathbf{s}_0$ and $\mathbf{s}_3$ (plotted with a diamond) maximizes $\mathbb{J}_{state}$, but the policy learned with this representation may not be optimal (as shown here). The original state representation (plotted with a star) is sufficient.

aliases all states, while the original state representation is at the far right (plotted with a star). The representation that aliases states $s_0$ and $s_3$ (plotted with a diamond) maximizes $\mathbb{J}_{state}$, but is insufficient to represent the optimal policy. Value iteration run with this state representation achieves only half the optimal return (0.5 versus 1.0).

### 5.3 Inverse Information

Since $\mathbb{J}_{inv}$ preserves state elements that the agent can influence with its actions, we might think that it is a good candidate for a sufficient objective that is capable of discarding more information than $\mathbb{J}_{fwd}$. However, here we show with a counterexample that $\mathbb{J}_{inv}$ is not sufficient. Intuitively, an insufficient representation can be obtained by maximizing $\mathbb{J}_{inv}$ in an MDP when the reward function depends on elements outside the agent's control. We then show that additionally requiring the representation to represent the immediate reward is not enough to resolve this issue.

**Proposition 3.** *$\mathbb{J}_{inv}$ is not sufficient for all reward functions. Additionally, adding $I(R_t; Z_t)$ to the objective does not make it sufficient.*

*Proof.* We show this by counter-example with the deterministic-transition MDP defined in Figure 3 (left). Consider the representation that aliases the states $s_0$ and $s_1$. This state representation wouldn't be sufficient for control because the same actions taken from these two states lead to different next states, which have different rewards ($a_0$ leads to the reward from $s_0$ while $a_1$ leads to the reward from $s_1$). However, this representation maximizes $\mathbb{J}_{inv}$ because, given each pair of states, the action is identifiable. Interestingly, this problem cannot be remedied by simply requiring that the representation also be capable of predicting the reward at each state. Indeed, the same insufficient representation from the above counterexample also maximizes this new objective as long as the reward at $s_0$ and $s_1$ are the same. $\square$

Analogous to the preceding section, in Figure 3 (right), we plot the values of the objectives $\mathbb{J}_{inv}$ and $\mathbb{J}_{fwd}$ for state representations ordered by increasing $I(Z; S)$ value (decreasing compression). The representation that aliases states $s_0$ and $s_1$ (plotted with a diamond) maximizes $\mathbb{J}_{inv}$, but is insufficient to represent the optimal policy; value iteration with this state representation achieves half the optimal return (0.5 versus 1.0).

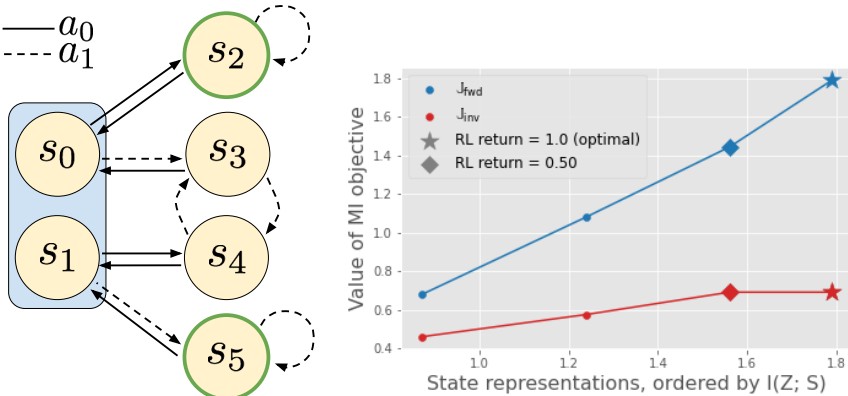

Figure 3: (left) In this MDP, a representation that aliases the states $s_0$ and $s_1$ into a single state maximizes $\mathbb{J}_{inv}$, yet is not sufficient to represent the optimal policy, which must distinguish between $s_0$ and $s_1$ in order to take a different action (towards the high-reward states outlined in green). (right) Values of $\mathbb{J}_{inv}$ and $\mathbb{J}_{fwd}$ for a few selected state representations, ordered by increasing $I(Z; S)$. The representation that aliases $s_0$ and $s_1$ (plotted with a diamond) maximizes $\mathbb{J}_{inv}$, but is not sufficient to learn the optimal policy. Note that this counterexample holds also for $\mathbb{J}_{inv} + I(R; Z)$.

# 6 Experiments

To analyze whether the conclusions of our theoretical analysis hold in practice, we present experiments studying MI-based representation learning with image observations. We do not aim to show that any particular method is necessarily better or worse, but rather to determine whether the sufficiency arguments that we presented can translate into quantifiable performance differences in deep RL.

## 6.1 Experimental Setup

To separate representation learning from RL, we first optimize each representation learning objective on a dataset of offline data, similar to the protocol in Stooke et al. [64]. Our datasets consist of 50k transitions collected from a uniform random policy, which is sufficient to cover the state space in our environments. We then freeze the weights of the state encoder learned in the first phase and train RL agents with the representation as state input. We perform our experiments on variations on the pygame [59] video game *catcher*, in which the agent controls a paddle that it can move back and forth to catch fruit that falls from the top of the screen (see

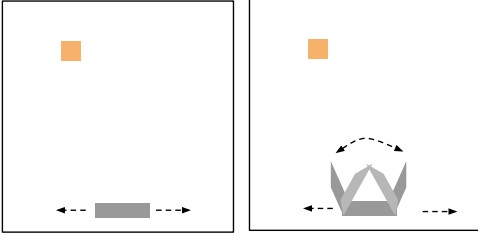

Figure 4: (left) Original *catcher* game: agent (grey paddle) moves left or right to catch fruit (yellow square) that falls. (right) Variation *catcher-grip*: agent must open the gripper to catch fruit.

Figure 4). A positive reward is given when the fruit is caught and a negative reward when the fruit is not caught. The episode terminates after one piece of fruit falls. We optimize $\mathbb{J}_{fwd}$ and $\mathbb{J}_{state}$ with noise contrastive estimation [24], and $\mathbb{J}_{inv}$ by training an inverse model via maximum likelihood. We also include the performance of an agent trained "end-to-end" with pixel inputs. For the RL algorithm, we use the Soft Actor-Critic algorithm [25], modified slightly for the discrete action distribution. Please see Appendix A.2 for full experimental details.

## 6.2 Computational Results

In principle, a representation learned with $\mathbb{J}_{inv}$ may not be sufficient to solve the *catcher* game. Because the agent does not control the fruit, a representation maximizing $\mathbb{J}_{inv}$ might discard that information, thereby making it impossible to represent the optimal policy. We observe in Figure 5 (top left) that indeed the representation trained to maximize $\mathbb{J}_{inv}$ results in RL agents that converge slower and to a lower asymptotic expected return. Further, attempting to learn a decoder from the learned representation to the position of the fruit incurs a high error (Figure 5, bottom left), indicating that the fruit is not precisely captured by the representation. The characteristics of this simulated game are representative of realistic tasks. Consider an autonomous vehicle that is stopped at a stoplight. Because the agent does not control the state of the stoplight, it may not be captured in the representation learned by $\mathbb{J}_{inv}$ and the resulting RL policy may choose to run the light.

In the second experiment, we consider a failure mode of $\mathbb{J}_{state}$. We augment the paddle with a gripper that the agent controls and must be open in order to properly catch the fruit (see Figure 4, right). Since the change in the gripper is completely controlled by a single action, the current state contains no information about the state of the gripper in the future. Therefore, a representation maximizing $\mathbb{J}_{state}$ might alias states where the gripper is open with states where the gripper is closed. Indeed, we see that the error in predicting the state of the gripper from the representation learned via $\mathbb{J}_{state}$ is about chance (Figure 5, bottom right). This degrades the performance of an RL agent trained with this state representation since the best the agent can do is move under the fruit and randomly open or close the gripper (Figure 5, top right). In the driving example, suppose turning on the headlights incurs positive reward if it's raining but negative reward if it's sunny. The representation could fail to distinguish the state of the headlights, making it impossible to learn when to properly use them. $\mathbb{J}_{fwd}$ produces useful representations in all cases, and is equally or more effective than learning representations purely from the RL objective alone (Figure 5).

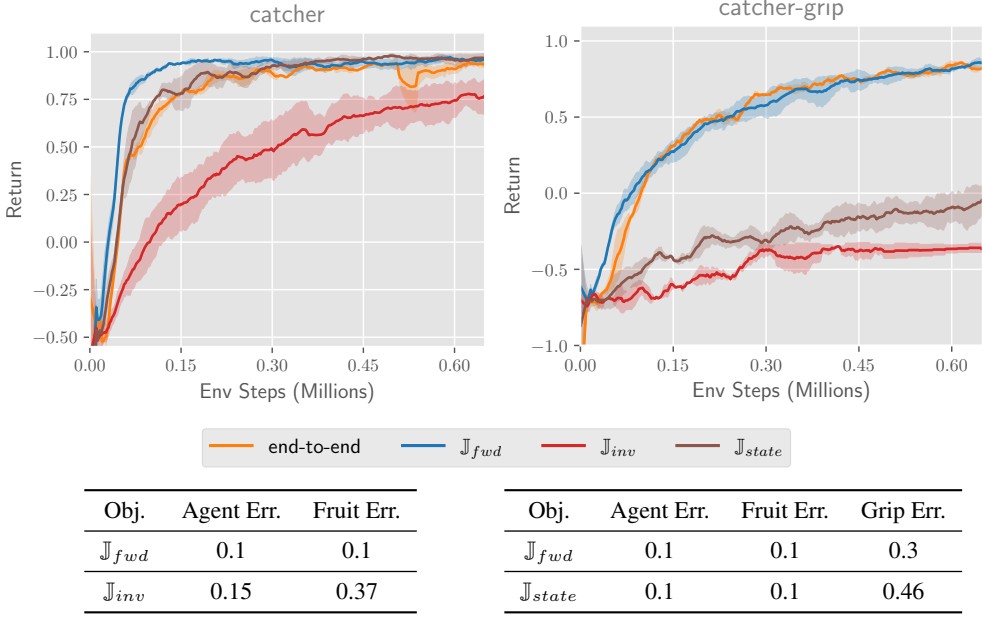

| Obj. | Agent Err. | Fruit Err. |
|------|-----------|-----------|
| $\mathbb{J}_{fwd}$ | 0.1 | 0.1 |
| $\mathbb{J}_{inv}$ | 0.15 | 0.37 |

| Obj. | Agent Err. | Fruit Err. | Grip Err. |
|------|-----------|-----------|-----------|
| $\mathbb{J}_{fwd}$ | 0.1 | 0.1 | 0.3 |
| $\mathbb{J}_{state}$ | 0.1 | 0.1 | 0.46 |

Figure 5: (left) Policy performance using learned representations as state inputs to RL, for the *catcher* and *catcher-grip* environments. (right) Error in predicting the positions of ground truth state elements from each learned representation. Representations maximizing $\mathbb{J}_{inv}$ need not represent the fruit, while representations maximizing $\mathbb{J}_{state}$ need not represent the gripper, leading these representations to perform poorly in *catcher* and *catcher-grip* respectively.

### 6.3 Increasing visual complexity via background distractors

Here we test whether sufficiency of representation can impact agent performance in more visually complex environments by adding background distractors to the agent's observations. We replace the background of the game with randomly generated images of 10 circles of different colors (Figure 6).

Analogous to Section 6, in Figure 7 we show the performance of an RL agent trained with the frozen representation as input (top), as well as the error of decoding true state elements from the representation (bottom). In both games, end-to-end RL from images performs poorly, demonstrating the need for representation learning. As predicted by the theory, the representation learned by $\mathbb{J}_{inv}$ fails in both games, and the representation learned by $\mathbb{J}_{state}$ fails in the *catcher-grip* game. The difference in performance between sufficient and insufficient objectives is even more pronounced in this setting. With more information present in the form of the distractors, insufficient objectives that do not optimize for representing all the required state

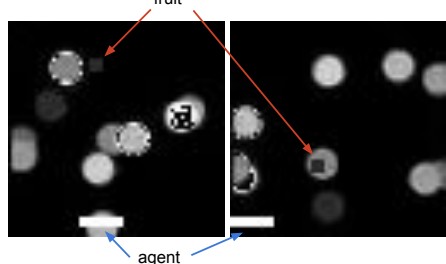

Figure 6: Example 64x64 pixel agent observations with background distractors (circles), randomly generated at each time step, that increase the difficulty of learning a good representation.

information may be "distracted" by representing the background objects instead, resulting in low performance. In Appendix A.5 we experiment with visual distractors that are temporally correlated across time. We also consider variations on our analysis, evaluating how well the representations predict the predict the optimal $Q^*$ in Appendix A.3, and experimenting with a different data distribution for collecting the representation learning dataset in Appendix A.4. These results demonstrate that using insufficient representation learning objectives *can* degrade the performance of an RL agent, not that they necessarily *will*. Our aim is to provide an illustration of the potential impact of insufficient representations, to underscore the utility of sufficiency as a tool in designing representation learning objectives and debugging system failures.

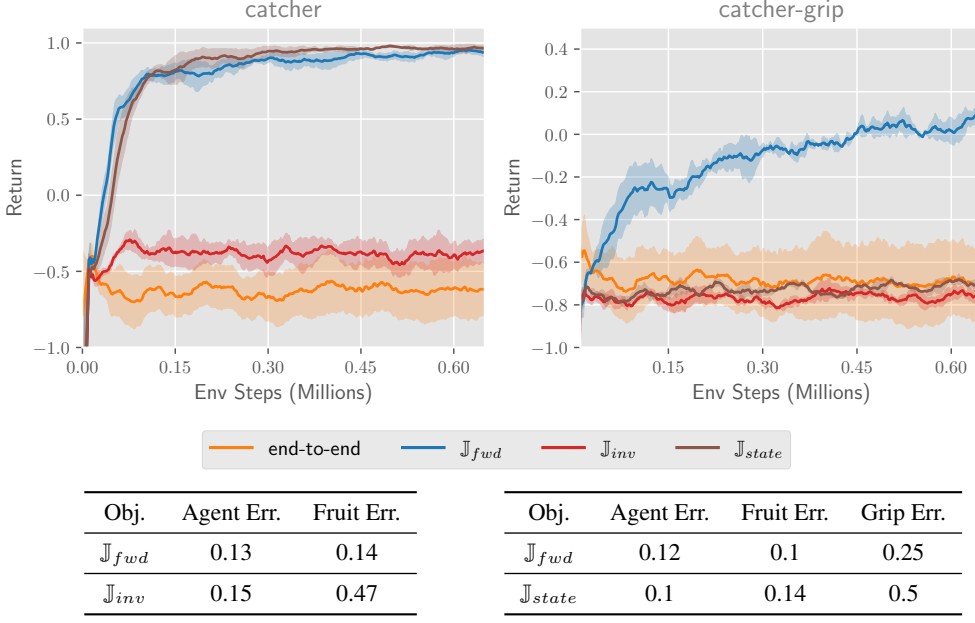

| Obj. | Agent Err. | Fruit Err. |
|------|-----------|-----------|
| $\mathbb{J}_{fwd}$ | 0.13 | 0.14 |
| $\mathbb{J}_{inv}$ | 0.15 | 0.47 |

| Obj. | Agent Err. | Fruit Err. | Grip Err. |
|------|-----------|-----------|-----------|
| $\mathbb{J}_{fwd}$ | 0.12 | 0.1 | 0.25 |
| $\mathbb{J}_{state}$ | 0.1 | 0.14 | 0.5 |

Figure 7: With background distractors added to the observations, the state representation learned via $\mathbb{J}_{inv}$ fails to capture the fruit object accurately in *catcher* (left), and the representation learned via $\mathbb{J}_{state}$ continue to perform poorly at capturing the gripper state in *catcher-grip* (right). The performance of the insufficient representations is even lower than in the clean background experiment.

## 7 Discussion

In this work, we analyze which common MI-based representation learning objectives are guaranteed to yield representations provably sufficient for learning the optimal policy. We show that two common objectives $\mathbb{J}_{state}$ and $\mathbb{J}_{inv}$ yield theoretically insufficient representations, and provide a proof of sufficiency for $\mathbb{J}_{fwd}$. We then show that insufficiency of representations can degrade the performance of deep RL agents with experiments on a simulated environment with visual observations. While an insufficient representation learning objective *can* work well for training RL agents on simulated benchmark environments, the same objective may fail for a real-world system with different characteristics. We believe that encouraging a focus on evaluating the sufficiency of newly proposed representation learning objectives can help better predict potential failures.

While sufficiency is a critical criterion for representations, compression is also highly important. Our results thus highlight an important open problem in unsupervised learning for RL: defining representation learning objectives that are provably sufficient *and* able to discard more information than $\mathbb{J}_{fwd}$. Is the "smallest" representation that maximizes $\mathbb{J}_{fwd}$ the smallest sufficient representation possible without making further assumptions on the MDP structure or reward function? What assumptions on the MDP structure or reward function would suffice to make $\mathbb{J}_{inv}$ and $\mathbb{J}_{state}$ sufficient? For example, $\mathbb{J}_{state}$ is trivially sufficient when the environment dynamics and the agent's policy are deterministic. However, we hypothesize there may be more interesting MDP classes, related to realistic applications, in which generally insufficient objectives may be sufficient.

Additionally, extending our analysis to the partially observed setting would be more reflective of practical applications. An interesting class of models to consider in this context are generative models such as variational auto-encoders [34]. Prior work has shown that maximizing the ELBO alone cannot control the content of the learned representation [28, 54, 3]. We conjecture that the zero-distortion maximizer of the ELBO would be sufficient, while other solutions would not necessarily be. We see these directions as fruitful in providing a deeper understanding of the learning dynamics of deep RL, and potentially yielding novel algorithms for provably accelerating RL with representation learning.

**Acknowledgements.** We would like to thank Ignasi Clavera, Chelsea Finn, and Ben Poole for insightful conversations at different stages of this work. This research was supported by the DARPA Assured Autonomy program and ARL DCIST CRA W911NF-17-2-0181.

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
