# A Appendix

## A.1 Sufficiency of $\mathbb{J}_{fwd}$: Proof of Proposition 1

We describe the proofs for the sufficiency results from Section 5 here. We begin by providing a set of lemmas, before proving the sufficiency of $\mathbb{J}_{fwd}$.

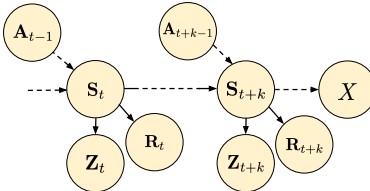

Figure 8: Graphical model for Lemma 1, depicting true states $S$, states in the representation $Z$, actions $A$, rewards $R$, and the variable $X$ (which we will interpret as the sum of future rewards in the proof of Proposition 1).

**Lemma 1.** *Let $X$ be a random variable dependent on $S_{t+k}$, with the conditional independence assumptions implied by the graphical model in Figure 8. (In the main proof of Proposition 1, we will let $X$ be the sum of rewards from time $t+k$ onwards.) If $I(Z_{t+k}; Z_t, A_t) = I(S_{t+k}; S_t, A_t) \forall k$, then $I(X; Z_t, A_t) = I(X; S_t, A_t) \forall k$.*

*Proof.* Recall from Definition 1 that $\phi_{\mathcal{Z}}(\mathbf{s}) = p(Z|S = \mathbf{s})$. For proof by contradiction, assume there is some $\phi_{\mathcal{Z}}$ and some $X$ such that $I(X; Z_t, A_t) < I(X; S_t, A_t)$ and that $I(Z_{t+k}; Z_t, A_t) = I(S_{t+k}; S_t, A_t)$. Note that for general variables $X, Y$, and $Z$, $X \to Y \to Z$ and $X \leftarrow Y \to Z$ have the same conditional independence relationships. Applying this fact to the structure $Z_t \leftarrow S_t \to S_{t+k} \to Z_{t+k}$ allows us to use the data processing inequality (DPI) to say that

$$I(Z_{t+k}; Z_t, A_t) \leq I(S_{t+k}; Z_t, A_t) \leq I(S_{t+k}; S_t, A_t) \tag{5}$$

We will proceed by showing the following, which gives the needed contradiction:

$$I(X; Z_t, A_t) < I(X; S_t, A_t) \implies I(S_{t+k}; Z_t, A_t) < I(S_{t+k}; S_t, A_t) \tag{6}$$

Using chain rule, we can expand the following expression in two different ways.

$$I(X; Z_t, S_t, A_t) = I(X; Z_t|S_t, A_t) + I(X; S_t, A_t) = 0 + I(X; S_t, A_t) \tag{7}$$

$$I(X; Z_t, S_t, A_t) = I(X; S_t|Z_t, A_t) + I(X; Z_t, A_t) \tag{8}$$

Note that the first term in Equation 7 is zero by the conditional independence assumptions in Figure 8. Equating the expansions, we can see that to satisfy our assumption that $I(X; Z_t, A_t) < I(X; S_t, A_t)$, we must have that $I(X; S_t|Z_t, A_t) > 0$.

Now we follow a similar procedure to expand the following expression:

$$I(S_{t+k}; Z_t, S_t, A_t) = I(S_{t+k}; Z_t|S_t, A_t) + I(S_{t+k}; S_t, A_t) = 0 + I(S_{t+k}; S_t, A_t) \tag{9}$$

$$I(S_{t+k}; Z_t, S_t, A_t) = I(S_{t+k}; S_t|Z_t, A_t) + I(S_{t+k}; Z_t, A_t) \tag{10}$$

The first term in Equation 9 is zero by the conditional independence assumptions in Figure 8. Comparing the first term in Equation 10 with the first term in Equation 8, we see that because $S_t \to S_{t+k} \to X$ form a Markov chain, by the DPI $I(S_{t+k}; S_t|Z_t, A_t) \geq I(X; S_t|Z_t, A_t)$. Therefore we must have $I(S_{t+k}; S_t|Z_t, A_t) > 0$. Combining Equations 9 and 10:

$$I(S_{t+k}; S_t, A_t) = I(S_{t+k}; S_t|Z_t, A_t) + I(S_{t+k}; Z_t, A_t) \tag{11}$$

Since $I(S_{t+k}; S_t|Z_t, A_t) > 0$, $I(S_{t+k}; Z_t, A_t) < I(S_{t+k}; S_t, A_t)$, which is exactly the contradiction we set out to show. $\square$

**Lemma 2.** *If $I(Y;Z) = I(Y;X)$ and $Y \perp Z | X$, then $\forall x, p(Y|X = x) = \mathbb{E}_{p(Z|X=x)} p(Y|Z)$.*

*Proof.* First note that the statement is not trivially true. Without any assumption regarding MI, we can write,

$$p(Y|X = x) = \int p(Y, Z|X = x) dz = \int p(Y|Z, X = x) p(Z|X = x) dz \tag{12}$$

Comparing this with the statement we'd like to prove, we can see that the key idea is to show that the MI equivalence implies that $p(Y|Z, X = x) = p(Y|Z)$.

Another way to write what we want to show is

$$D_{KL}[p(Y|X)||\mathbb{E}_{p(Z|X)} p(Y|Z)] = 0 \tag{13}$$

Our strategy will be to upper-bound this KL by a quantity that we will show to be 0. Since KL divergences are always lower-bounded 0, this will prove equality. We begin by writing out the definition of the KL divergence, and then using Jensen's inequality to upper-bound it.

$$
\begin{aligned}
D_{KL}[p(Y|X)||\mathbb{E}_{p(Z|X)} p(Y|Z)] &= \mathbb{E}_{p(Y|X)} \log \left[ \frac{p(Y|X)}{\mathbb{E}_{p(Z|X)} p(Y|Z)} \right] \\
&= \mathbb{E}_{p(Y|X)} [\log p(Y|X) - \log \mathbb{E}_{p(Z|X)} p(Y|Z)] \\
&\leq \mathbb{E}_{p(Y|X)} [\log p(Y|X) - \mathbb{E}_{p(Z|X)} \log p(Y|Z)]
\end{aligned}
\tag{14}
$$

where the last inequality follows by Jensen's inequality. Because KL divergences are greater than 0, this last expression is also greater than 0. Now let us try to relate $I(Y;Z) = I(Y;X)$ to this expression. We can re-write this equality using the entropy definition of MI.

$$H(Y) - H(Y|Z) = H(Y) - H(Y|X) \tag{15}$$

The $H(Y)$ cancel and substituting the definition of entropy we have:

$$\mathbb{E}_{p(Y,Z)} \log p(Y|Z) = \mathbb{E}_{p(Y,X)} \log p(Y|X) \tag{16}$$

On the right-hand side, we can use the Tower property to re-write the expectation as

$$\mathbb{E}_{p(Y,X)} \log p(Y|X) = \mathbb{E}_{p(Z)} \mathbb{E}_{p(Y,X|Z)} \log p(Y|X) = \mathbb{E}_{p(X)p(Y|X)p(Z|X)} \log p(Y|X) \tag{17}$$

Similarly, we can re-write the left-hand side as

$$\mathbb{E}_{p(Y,Z)} \log p(Y|Z) = \mathbb{E}_{p(X)} \mathbb{E}_{p(Y,Z|X)} \log p(Y|Z) = \mathbb{E}_{p(X)p(Y|X)p(Z|X)} \log p(Y|Z) \tag{18}$$

Subtracting the right-hand side from the left-hand side, we have

$$
\begin{aligned}
\mathbb{E}_{p(X)p(Y|X)p(Z|X)} [\log p(Y|X) - \log p(Y|Z)] &= 0 \\
\mathbb{E}_{p(X)} [\mathbb{E}_{p(Y|X)} [\log p(Y|X) - \mathbb{E}_{p(Z|X)} \log p(Y|Z)]] &= 0
\end{aligned}
\tag{19}
$$

Equation 14 tells us that the term inside the expectation $\mathbb{E}_{p(X)}$ is greater than 0. Equation 19 tells us that the expectation of this term is equal to 0. If the sum of elements that all have the same sign is zero, then each element is zero. Therefore,

$$0 = \mathbb{E}_{p(Y|X)} [\log p(Y|X) - \mathbb{E}_{p(Z|X)} \log p(Y|Z)] \geq D_{KL}[p(Y|X)||\mathbb{E}_{p(Z|X)} p(Y|Z)] \tag{20}$$

Since the KL divergence is upper and lower-bounded by 0, it must be equal to 0, and therefore $p(Y|X) = \mathbb{E}_{p(Y|Z)} p(Z|X)$ as we wanted to show. $\qquad \square$

Given the lemmas stated above, we can then use them to prove the sufficiency of $\mathbb{J}_{fwd}$.

**Proposition 1.** *(Sufficiency of $\mathbb{J}_{fwd}$) Let $(\mathcal{S}, \mathcal{A}, \mathcal{T})$ be an MDP with dynamics $p(S_{t+1}|S_t, A_t)$. Let the policy distribution $p(A|S)$ and steady-state state occupancy $p(S)$ have full support on the action and state alphabets $\mathcal{A}$ and $\mathcal{S}$ respectively. Let $\mathcal{R}$ be the set of all reward functions that can be expressed as a function of the state, $r : \mathcal{S} \to \mathbb{R}$. [3] See Figure 8 for a graphical depiction of the conditional independence relationships between variables.*

*For a representation $\phi_{\mathcal{Z}}$, if $I(Z_{t+1}; Z_t, A_t)$ is maximized $\forall t > 0$ then $\forall r \in \mathcal{R}$ and $\forall \mathbf{s}_1, \mathbf{s}_2 \in \mathcal{S}$, $\phi_{\mathcal{Z}}(\mathbf{s}_1) = \phi_{\mathcal{Z}}(\mathbf{s}_2) \implies \forall \mathbf{a}, Q_r^*(\mathbf{s}_1, \mathbf{a}) = Q_r^*(\mathbf{s}_2, \mathbf{a})$.*

*Proof.* Note that $(Z_{t+1}; Z_t, A_t)$ is maximized if the representation $\phi_{\mathcal{Z}}$ is taken to be the identity. In other words $\max_\phi I(Z_{t+1}; Z_t, A_t) = I(S_{t+1}; S_t, A_t)$.

Define the random variable $\bar{R}_t$ to be the discounted return starting from state $\mathbf{s}_t$.

$$\bar{R}_t = \sum_{i=1}^{H-t} \gamma^k R_{t+i} \tag{21}$$

Plug in $\bar{R}_t$ for the random variable $X$ in Lemma 1:

$$I(Z_{t+1}; Z_t, A_t) = I(S_{t+1}; S_t, A_t) \quad \implies \quad I(\bar{R}_{t+1}; Z_t, A_t) = I(\bar{R}_{t+1}; S_t, A_t) \tag{22}$$

Now let $X = [S_t, A_t]$, $Y = \bar{R}_t$, and $Z = Z_t$, and note that by the structure of the graphical model in Figure 8, $Y \perp Z | X$. Plugging into Lemma 2:

$$\mathbb{E}_{p(\mathbf{z}_t|S_t=\mathbf{s})} p(\bar{R}_t|Z_t, A_t) = p(\bar{R}_t|S_t = \mathbf{s}, A_t) \tag{23}$$

Now the $Q$-function given a reward function $r$ and a state-action pair $(\mathbf{s}, \mathbf{a})$ can be written as an expectation of this random variable $\bar{R}_t$, given $S_t = \mathbf{s}$ and $A = \mathbf{a}$. (Note that $p(\bar{R}_t|S_t = \mathbf{s}, A_t = \mathbf{a})$ can be calculated from the dynamics, policy, and reward distributions.)

$$Q_r(\mathbf{s}, \mathbf{a}) = \mathbb{E}_{p(\bar{R}_t|S_t=\mathbf{s}, A_t=\mathbf{a})}[\bar{R}_t] \tag{24}$$

Since $\phi_{\mathcal{Z}}(\mathbf{s}_1) = \phi_{\mathcal{Z}}(\mathbf{s}_2)$, $p(\mathbf{z}_t|S_t = \mathbf{s}_1) = p(\mathbf{z}_t|S_t = \mathbf{s}_2)$. Therefore by Equation 23, $p(\bar{R}_t|S_t = \mathbf{s}_1, A_t) = p(\bar{R}_t|S_t = \mathbf{s}_2, A_t)$. Plugging this result into Equation 24, $Q_r(\mathbf{s}_1, \mathbf{a}) = Q_r(\mathbf{s}_2, \mathbf{a})$. Because this reasoning holds for all $Q$-functions [4], it also holds for the optimal $Q$, therefore $Q_r^*(\mathbf{s}_1, \mathbf{a}) = Q_r^*(\mathbf{s}_2, \mathbf{a})$.

$\square$

---

[3] If the reward is a function of the action, then $\mathbb{J}_{fwd}$ may not be sufficient. To see this, first consider an MDP where the reward depends on a random variable in the state that is independently sampled at each timestep. In this case, $\mathbb{J}_{fwd}$ is still sufficient, because all policies will be equally poor since the reward at the next timestep is completely unpredictable. However, if the reward depends on the *action* taken, then a policy trained on the original state space could choose the high-reward action, but a representation learned by $\mathbb{J}_{fwd}$ needn't capture that information if it doesn't improve prediction of the next state. So in this latter case, $\mathbb{J}_{fwd}$ can be insufficient.

[4] Note this result is stronger than what we needed: it means that representations that maximize $\mathbb{J}_{fwd}$ are guaranteed to be able to represent even sub-optimal $Q$-functions. This makes sense in light of the fact that the proof holds for all reward functions - the sub-optimal $Q$ under one reward is the optimal $Q$ under another.

## A.2 Experimental Details and Further Experiments

### A.2.1 Didactic Experiments

The didactic examples are computed as follows. Given the list of states in the MDP, we compute the possible representations, restricting our focus to representations that group states into "blocks." We do this because there are infinite stochastic representations and the MI expressions we consider are not convex in the parameters of $p(Z|S)$, making searching over these representations difficult. Given each state representation, we compute the value of the MI objective as well as the optimal value function using exact value iteration. In these examples, we assume that the policy distribution is uniform, and that the environment dynamics are deterministic. Since we consider the infinite horizon setting, we use the steady-state state occupancy in our calculations.

**Example manual MI calculation**. As an example, we calculate the value of $\mathbb{J}_{state}$ manually for both the identity representation and the one that aliases $s_0$ and $s_3$ (these correspond to the representations plotted with diamond and star respectively in the plot on the right side of Figure 2). We can expand MI in terms of entropies: $\mathbb{J}_{state} = I(Z'; Z) = H(Z') - H(Z'|Z)$.

For the identity representation, $Z = S$. The steady-state occupancy distribution is $p(S) = \frac{1}{4}$, so $H(S) = 4 * \sum \frac{1}{4} \log(4) = \log(4) = 2\log(2)$. We assume a uniform policy distribution, so $p(S'|S) = \frac{1}{2}$, and therefore $H(S'|S) = -8 * \frac{1}{2} * \frac{1}{4} * \log(\frac{1}{2}) = \log(2)$. Subtracting the two values, we get $\mathbb{J}_{state} = \log(2)$ for the identity representation.

For the representation that aliases $s_0$ and $s_3$: let us call the aliased state $z_0$. The steady-state occupancy is $p(Z = s_1) = p(Z = s_2) = \frac{1}{4}$ and $p(Z = z_0) = \frac{1}{2}$. Therefore $H(Z) = \frac{3}{2}\log(2)$. The conditional distributions are $p(Z = s_2|z_0) = p(Z = s_1|z_0) = \frac{1}{2}$, while $p(Z = z_0|s_1) = p(Z = z_0|s_2) = 1$. Therefore $H(Z'|Z) = -2 * (1) * \frac{1}{4}\log(1) - 2 * \frac{1}{2} * \frac{1}{2} * \log(\frac{1}{2}) = \frac{1}{2}\log(2)$. Subtracting the second term from the first, we see that for this representation as well, $\mathbb{J}_{state} = \log(2)$. Since both of these representations have the same value of $\mathbb{J}_{state}$, an algorithm maximizing $\mathbb{J}_{state}$ could not distinguish between them, potentially yielding the insufficient representation that aliases $s_0$ and $s_3$.

### A.2.2 Deep RL Experiments

The deep RL experiments with the catcher game are conducted as follows. First, we use a uniform random policy to collect 50k transitions in the environment. In this simple environment, the uniform random policy suffices to visit all states (the random agent is capable of accidentally catching the fruit, for example). Next, each representation learning objective is maximized on this dataset. For all objectives, the images are pre-processed in the same manner (resized to 64x64 pixels and normalized) and embedded with a convolutional network. The convolutional encoder consists of five convolutional layers with ReLU activations and produces a latent vector with dimension 256. We use the latent vector to estimate each mutual information objective, as described below.

**Inverse information**: We interpret the latent embeddings of the images $S_t$ and $S_{t+1}$ as the parameters of Gaussian distributions $p(Z|S_t)$ and $p(Z|S_{t+1})$. We obtain a single sample from each of these two distributions, concatenate them and pass them through a single linear layer to predict the action. The objective we maximize is the cross-entropy of the predicted actions with the true actions, as in Agrawal et al. 2016 and Shelhamer et al. 2016. To prevent recovering the trivial solution of preserving all the information in the image, we add an information bottleneck to the image embeddings. We tune the Lagrange multiplier on this bottleneck such that the action prediction loss remains the same value as when trained without the bottleneck. This approximates the objective $\min_\phi I(Z; S) s.t. I_{inv} = \max I_{inv}$. To use the learned encoder for RL, we embed the image from the current timestep and take the mean of the predicted distribution as the state for the RL agent.

**State-only information**: We follow the Noise Contrastive Estimation (NCE) approach presented in CPC (Oord et al. 2018). Denoting $Z_t$ and $Zt + 1$ as the latent embedding vectors from the convolutional encoders, we use a log-bilinear model as in CPC to compute the score: $f(Z_t, Z_{t+1}) = \exp(Z_t^T W Z_{t+1})$ for the cross-entropy loss. We also add an information bottleneck as described above, though we found that it actually isn't needed to obtain insufficient representations. To use the learned encoder for RL, we embed the image from the current timestep and use this latent vector as the state for the RL agent.

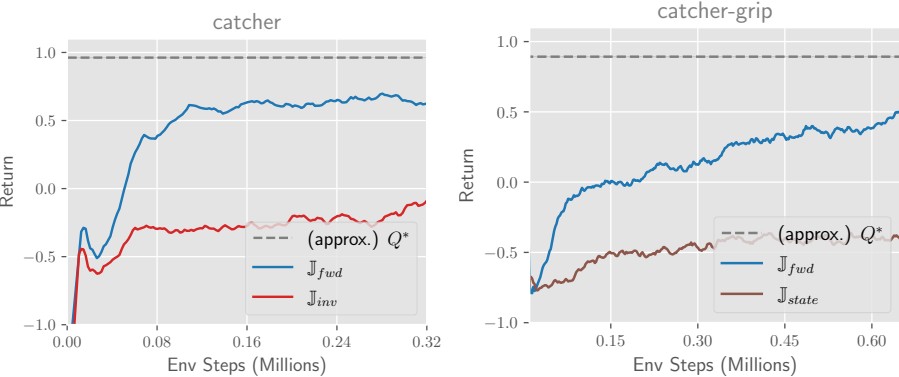

Figure 9: Policy performance obtained from $Q$-functions trained to predict $Q^*$ given state representations learned by each MI objective. Insufficient objectives $\mathbb{J}_{inv}$ and $\mathbb{J}_{state}$ respectively perform worse than sufficient objective $\mathbb{J}_{fwd}$.

**Forward information**: We follow the same NCE strategy as for state-only information, with the difference that we concatenate the action to $Z_t$ before computing the contrastive loss. We do apply the information bottleneck here since we want to be sure to evaluate the most compressed representation that maximizes $\mathbb{J}_{fwd}$.

We then freeze the state encoder learned via MI-maximization and use the representation as the state input for RL. The RL agent is trained using the Soft Actor-Critic algorithm [25], modified for the discrete action distribution (the Q-function outputs Q-values for all actions rather than taking action as input, the policy outputs the action distribution directly rather than parameters, and we can directly compute the expectation in the critic loss rather than sampling). The policy and critic networks consist of two hidden linear layers of 200 units each with ReLU activations. We run 5 random seeds of each experiment and plot the mean as a solid line and one standard deviation as a shaded region.

Our implementation for the deep RL experiments is based on Garage [20] and we use the default hyperparameters for the SAC algorithm found in that repository. Garage is distributed under the MIT License. The environments we use are from the pygame library [59] which is distributed under GNU LGPL version 2.1. We used NVIDIA Titan X graphics cards to accelerate the training of the representation learning algorithms with image inputs.

### A.3   Alternative evaluation of the representation by predicting $Q^*$

In Section 6, we evaluated the learned representations by running a temporal difference RL algorithm with the representation as the state input. In this section, instead of using the bootstrap to learn the $Q$-function, we instead regress the $Q$-function to the optimal $Q^*$. To do this, we first compute the (roughly) optimal $Q^*$ by running RL with ground truth game state as input and taking the learned $Q$ as $Q^*$. Then, we instantiate a new RL agent and train it with the learned image representation as input, regressing the $Q$-function directly onto the values of $Q^*$. We evaluate the policy derived from this new $Q$-function, and plot the results for both the *catcher* and *catcher-grip* environments in Figure 9. We find that similar to the result achieved using the bootstrap, the policy performs poorly when using representations learned by insufficient objectives ($\mathbb{J}_{inv}$ in *catcher* and $\mathbb{J}_{state}$ in *catcher-grip*). Interestingly, we find that the error between the learned $Q$-values and the $Q^*$-values is roughly the same for sufficient and insufficient representations. We hypothesize that this discrepancy between $Q$-value error and policy performance is due to the fact that small differences in $Q$-values on a small set of states can result in significant behavior differences in the policy.

### A.4   Changing the data distribution used for representation learning

In Section 6, we collect the dataset used to optimize the representation learning objectives with a uniform random policy. Here, we perform an ablation where we collect this dataset using the optimal policy, trained from ground truth state (agent and fruit positions). We show the results for the *catcher*

environment in Figure 10. As predicted by our theoretical analysis, $\mathbb{J}_{inv}$ is still insufficient, and can fail to represent the fruit, leading to poor performance of the RL agent.

## A.5 Temporally correlated visual distractors

In this section, we perform experiments with temporally correlated background distractors by animating the colored circles (see Figure 6) to bounce around the screen. We show results for the *catcher* environment in Figure 11. The results are similar to those in Figure 7 of the paper - the insufficient objective $\mathbb{J}_{inv}$ results in representations poorly suited to learning the task, while $\mathbb{J}_{fwd}$ yields a useful representation. However, the $\mathbb{J}_{fwd}$ objective could suffer if the complexity of the background were further increased. With limited model capacity, $\mathbb{J}_{fwd}$ might fixate on the moving background rather than the task-relevant fruit, because it is incentivised to predict all the state information that is predictable. Regardless of the background complexity, $\mathbb{J}_{inv}$ is still insufficient in this MDP.

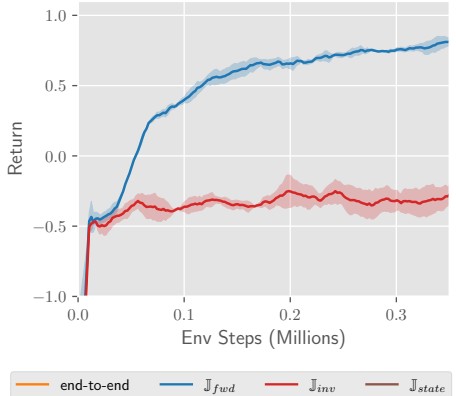

Figure 10: Performance of RL agents trained with state representations learned with data collected by the optimal policy, *catcher* environment.

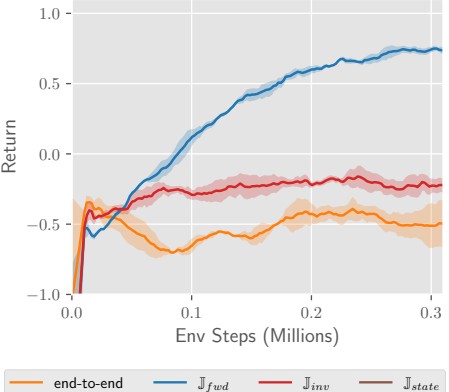

Figure 11: Performance of RL agents learned from state representations when observations contain visual distractors that are correlated across timesteps, *catcher* environment.