# OpenReview forum: "Which Mutual-Information Representation Learning Objectives are Sufficient for Control?"
_NeurIPS.cc/2021/Conference — NeurIPS 2021 Poster_

### Official Review · Reviewer_nwZo · 2021-07-16

**Rating:** 6
**Confidence:** 5

**Summary:**

The paper discusses on a theoretical level what MI-related quantities are well-suited for representation learning in the context of reinforcement learning. The paper argues that for reinforcement learning, forward information is the only one that is well suited for learning representations by a certain sufficiency definition, and state-only transition information and inverse information do not satisfy the sufficiency property; thus only the forward information is a well-suited principle for representation learning in reinforcement learning. Empirical results on simple offline data suggest the usefulness of the idea.

**Ethical Concerns:**

No.

**Limitations And Societal Impact:**

Yes.

**Main Review:**

**Update**: I have read the rebuttal. The authors clarified my confusion regarding the correctness of the proofs. I am willing to raise my score to a weak accept. I believe it is crucial to clarify the scope of the statements that are made in the paper, that is "reward depends only on the state" (otherwise the needed representations can be quite different). In retrospect, the choice of the word "control" in the title (and the graphical model in the appendix) might have implied this, but other notations (such as $R(s, a)$) could lead to some confusion.


-----

**Strengths**

I think this paper is an interesting read, and that it is justified to consider a very simplified view over this setting (mutual information and be well approximated from samples, exploration is a non-issue, etc.). If the theory is true, it can be used to discuss which MI-based objective is adequate for representation learning in RL. The distractor experiment is an interesting setting where there is an incentive to drop this "useless" information (since it cannot be predicted better than random, there is not much that the policy can do about it).

**Weaknesses**: Unfortunately, I think the proof of Proposition 1 is false as it is based on an not always true assumption from Lemma 2.

To make my argument, think about the case where the optimal mutual information in J_fwd is simply zero. For example, each episode is just a sequence of independent contextual bandit problems. So S_{t+1} is independent of S_{t} and A_{t} (which only determines R_t in contextual bandits), mutual information should be zero. Does it seem that even J_fwd would not be incentivized to learn a meaningful representation at all? To have a good representation, at least it seems that the reward should be taken into account?

Let us say in this case, we use the same representation for every state (giving us a mutual information of zero for J_fwd, and that is fine because each timestep is independent anyways now), and it seems that we should obviously not expect Q(s_1, a) = Q(s_2, a) to work for every pair of states? Clearly, it is not Q-sufficient?

This makes me wonder if there is something wrong with the proposition, and I believe is because you applied Lemma 2 where the assumption for Lemma 2 might not hold. In the proof, you use X = [S_t, A_t], Y = [R_t], Z = [Z_t]. To use Lemma 2, you would have to assume I(X; Y) = I(Y; Z), which is true if Z_t = (S_t, A_t), but clearly not true in the above example (where I(Z_t; R_t) = 0, and I(R_t; S_t, A_t) is not zero). So everything falls apart from here.

**Experiments**
In the experiments, the mutual information is estimated by InfoNCE, which is known to have a high bias when the batch size is small [Poole et al., ICML 2019]. How accurate is your estimated mutual information? I don't think I see enough details in the appendix to gauge this. This could be an important factor, because it would be natural to believe that the forward mutual information would be quite high in deterministic dynamics, where $Z_{t+1}$ is known once $Z_{t}$ and $A_t$ are known. I am curious about this in the catcher environment.

Is there any insight as to why J_state works poorly for catcher-grip despite working decently for catcher?



**Comments**

Statement of Lemma 2: the way it is written now makes it very easy to misunderstand as a "type-check" error. I think a more appropriate quantity on the rhs should be $E_{p(Z | X = x)}[p(Y | Z)]$, then one in Eq 11?



The type of Q functions are not consistent, sometimes it is $(s, a)$, others it is $(a, s)$. This needs to be fixed.

**Time Spent Reviewing:**

3 hours

---

> ### Author Response · Authors · 2021-08-06
> **Prop1 clarification; accuracy of estimated MI; insufficiency of state information for catcher-grip environment**
>
> Thank you for your careful reading of our paper! We respond to your questions below.
>
> “Question about proof of Prop. 1”: We believe that the proof of Proposition 1 is correct, and that the proposed contextual bandit example violates an explicit assumption stated in the paper. Note that in our setting, we assume that the reward is a function solely of the current state, and not the action (see the footnote on page 2 of the appendix). The independent contextual bandit example violates this assumption since the reward depends on the action taken. You are correct that if the reward could depend on the action, the representation would need to also be capable of predicting the current reward in order to be sufficient (which could be achieved for example by adding a reward prediction loss). Our assumption explicitly excludes this case.
>
> “Assumption for Lemma 2 might not hold”: We believe the assumption required for Lemma 2 does hold, as long as the reward is a function only of the current state as stated above. Note that when we apply Lemma 2, $Y = [\bar{R}_t]$ is the sum of future rewards, not including the reward at the current timestep. We prove in Lemma 1 that $I(X; Y) = I(Z; Y)$, where $X = [S_t, A_t], Y = [\bar{R}_t]$, and $Z = [Z_t]$.
>
> “Accuracy of estimated MI”: Unfortunately, estimating MI in high dimensions is very difficult. Prior work has shown that commonly used estimators fail to predict MI accurately in simple settings (Song & Ermon 2020). We make use of the NCE method that is commonly used to maximize MI objectives for representation learning in practice.
>
> “Why does $J_{state}$ work poorly for catcher-grip”: $J_{state}$ works poorly for catcher-grip because it is insufficient in this MDP. The intuition is that since the change in the gripper is completely controlled by a single action, the current state does not contain information about the state of the gripper in the future. Therefore, a representation maximizing $J_{state}$ might alias states where the gripper is open with states where the gripper is closed. Please refer to Lines 273-284 in Section 6.2 for a discussion.
>
> “Typos”: Thank you for pointing out the typo in the statement of Lemma 2. The RHS should be $E_{p(Z | X=x)} \[p(Y|Z)\]$. We will fix the $Q(a, s)$ typos as well.

---

> > ### Comment · Reviewer_nwZo · 2021-08-06
> > **Thanks for the clarification**
> >
> > Thanks for clarifying the confusion regarding the theory. I think with the additional assumption, the theory should be correct. I will update the review later.
> >
> > However, I feel like the fact that "reward only depends on current state" is very important and should be highlighted more than just a footnote. There are several ways to make it clear (since it is quite important).
> >
> > 1. Have a separate assumption statement at the beginning.
> > 2. Use R(s) instead of R(s, a) to cement this claim.
> > 3. In the proof of proposition 1, when you use Lemma 2, add some statements that show the application of Lemma 2 here is valid.

---

> > > ### Author Response · Authors · 2021-08-09
> > > **Thanks, we will incorporate your suggestions!**
> > >
> > > Thanks for these suggestions on how to make the assumption that "reward depends on current state" more clear; we agree they would improve the clarity of the paper, and we will add them.

---

### Official Review · Reviewer_qUv5 · 2021-07-17

**Rating:** 6
**Confidence:** 4

**Summary:**

This paper analyzes three mutual information-based objectives in representation learning for RL from the perspective of sufficiency. The main contribution of this work is to theoretically prove that the forward information metric exhibits, while two other metrics, state-only transition information and inverse information, are not sufficient. Giving numerical examples demonstrating the impact of insufficiency on the RL performance is also a meaningful result presented in this work.

**Limitations And Societal Impact:**

- While the relation between the sufficiency and the learning performance is demonstrated in the presented example, how much generalizable this observation is not clearly explained. It would be good, if the authors would be able to give more systematic explanation on what classes of RL problems the notion of sufficiency makes difference.
- In the example with distraction, it is not clearly explained why the end-to-end RL does not perform well. It may not be the case that the end-to-end approach has a fundamental limitation in representation, as then the first example would look different. That said, the limited performance of the end-to-end scheme is due to rather practical/numerical issues. This may be okay, but this could bring some question if such a practical issue would affect the other representations. In other words, how could the other reasons be discarded that may affect the performance degradation of insufficient mutual information metrics?
- The result of this paper reads like we should consider the forward information metric. Would there be any disadvantages and/or costs of this metric compared to the other two? Is the takeaway that we should use the forward information, or we need to be more careful about the choice of metric? Such a discussion is needed in this paper.

**Main Review:**

-  ​The presented theoretical analysis seems valid and correct. While the sufficiency/insufficiency results themselves are, in some sense, intuitively straightforward, the theoretical proof for the sufficiency of forward information and the counterexamples for the other two metrics are very interesting.
-This paper is generally well-written, delivering the main message clearly. However, the reviewer suggests to include more details of the proof of proposition 1 in the main text rather than appendix.
- Given that many recent RL schemes use various mutual information metric, the theoretical findings in this paper would provide some insight on how these work would be improved and enhanced.


**Time Spent Reviewing:**

4 hours

---

> ### Author Response · Authors · 2021-08-06
> **Generalizability of sufficiency analysis; why end-to-end RL fails with distractors; disadvantages of forward information**
>
> Thank you for your thoughtful review.
>
> *Response to limitations*:
>
> “Generalizability of sufficiency analysis”: The sufficiency analysis for $J_{fwd}$ applies across all MDPs which satisfy the assumptions stated in Proposition 1, while the counterexamples simply suggest that for $J_{inv}$ and $J_{state}$, there *exist* MDPs where they are insufficient.  If the state representation is not sufficient, then it may be impossible to learn and represent optimal policies and Q-functions, making the performance of the resulting RL agent very poor.
>
> “Why does end-to-end RL fail in environment with distractors?”: In end-to-end deep RL, representation learning is entangled with RL optimization. Therefore, it’s difficult to say definitively that the end-to-end RL baseline fails due to representation learning difficulties, but we have some evidence to believe this is the case: (1) RL from the ground truth state representation performs well, (2) Regression from the images with distractors to the ground truth state achieves low error.
>
> “Disadvantages of the forward information objective”: We discuss the shortcomings of $J_{fwd}$ in Section 7, where we note that this objective lacks a notion of “task-relevance”, and so may capture much more information than is required if the reward function is known. We would emphasize that our aim is not to suggest that the forward information objective is in some way an “ideal” objective to use for representation learning, but rather to provide an analysis of which existing representation learning objectives are sufficient. Of course, other characteristics are also desirable in a good representation, such as being disentangled and minimal. We will qualify our discussion more carefully in this regard in the text.

---

> > ### Comment · Reviewer_qUv5 · 2021-08-18
> > **Author response makes general sense**
> >
> > The authors' response has sufficiently clarified key issues the reviewer raised. The reviewer thinks that it is important this clarification is clearly delivered in the manuscript as well.

---

### Official Review · Reviewer_Ncmr · 2021-07-26

**Rating:** 6
**Confidence:** 4

**Summary:**

The paper investigates three popular mutual information (MI) based objectives for learning reinforcement learning (RL) representations through the lens of sufficiency. The authors distinguish between two types of sufficiency: (i) a state representation is $\pi^*$-sufficient if it can be used to represent the optimal policy; and (ii) it is $Q^*$-sufficient if it can additionally guarantee learnability (i.e., convergence of Q-learning to the optimal policy with that representation). The paper argues that the latter is the right type of sufficiency, and show that only one out of the three popular MI-based objectives satisfies is sufficient. For the other two, they provide counterexamples illustrating simple conditions under which these objectives can fail. The authors additionally verify their claims via deep RL experiments on variations of a video game.

**Limitations And Societal Impact:**

The author cover the societal impact by providing citations.

**Main Review:**

There are several strong points about this work.
- It investigates a very important question. Various MI-based objectives for state representation have been previously proposed in the literature. A straightforward way to compare the different objectives is via empirical comparison on various RL environments. In addition to that though, we also need formalisms that compare the objectives along rigorous criteria. The authors introduce such a formalism based on the concept of sufficiency. This is a concrete way to compare the resulting representations, with a strong intuitive basis: at the very least, we want representations that can guarantee learnability.
- The paper contains several original elements, including the sufficiency proof for the forward information objective, and the illuminating counterexamples for the other two objectives.
- The experiments are smartly designed to make the insufficient objectives fail by
- The paper is clearly written, and simple to follow. I believe that the proofs are technically sound, except one detail in Lemma 1 that I am not so clear about (see below).
- The related work is covered well. Furthermore, the discussion section contains many interesting points.

My concerns are as follows.
- The authors do an excellent job in highlighting when the state only MI and inverse MI can fail under the lens of sufficiency. Their experiments provide further evidence for that. However, it is far from clear whether the notion of sufficiency is really the right formalism to compare different objectives in practice. The experiments are rather artificial and specifically designed to showcase specific weaknesses of the two objectives without sufficiency. I think a reasonable way to understand whether sufficiency is really important in practice would be to compare empirical performance of the various objectives in many benchmarked problems. If the two objectives without the sufficiency property suffer from poor performance often, this would indicate that the lack of sufficiency makes a big impact in practice. If not, this could suggest that in practice the situation is more complex.
- Related to the point above, it almost appears after reading this work that the forward objective is the clear winner. But is this always the case though in practical settings? One obvious problem with the forward objective is that it does not consider information about the reward, only about the state and action. Furthermore, it lacks a notion of task relevance, as the authors point out. Conversely, what would be the strengths of the state only and inverse objectives, compared to the forward objective. I am getting the feeling that the comparison is kind of one-sided.
- As the authors point out, the notion of sufficiency that is central in their paper has in fact been explored in several past works on state abstraction (e.g., Li, Walsh and Littman 2006). Even though it is true that this work provides a formal framework for comparing the various objectives, the central ideas behind this have already been laid out in previous works.

Additional questions to authors.
- What happens when we combine the various objectives? We could for example consider some weighted combination of the different objectives. One might wonder whether there is any benefit when learning representations that take into account multiple MI objectives. Would that help performance?
- It makes sense why in general we want a uniform random policy to collect data to learn the state representation: this will allow us to cover the entire state space well. I was wondering though: for the forward objective would it help if we collect the data to learn the state representation via standard RL (i.e., via an evolving policy)? If the RL policy only visits a small part of the state space, then this could allow us to learn even smaller representations Z_t, as we would not need to be equally predictive of all predictable elements in all states but only in the states that matter in the RL policy. This may also be much easier than collecting data with a random uniform policy, especially in very high-D spaces.
- In Lemma 1, the authors claim that Z_t->S_t->S_{t+k}->Z_{t+k} form a Markov chain. In the diagram, we see that the dependency is S_t->Z_t. But in the above claim the direction is reversed. Is this valid, and if yes, why? I was thinking that the direction of the arrow could make a difference. For example, assume Y<-X->Z, and consider the trivial case when Y=Z (and both depend on X). In that case, it can be that I(Y;Z)>I(Y;X)=I(Z;X). But if we reverse the arrow, we get the Markov chain Y->X->Z. In that case, the data processing inequality would imply that I(Y;X)>=I(Y;Z), which is a contradiction. This may mean that we cannot always reverse the arrow. Perhaps I am missing something? I believe the claim in Lemma 1 is still correct, but I was wondering about that part of the proof.
-- The forward objective is trivially maximized if Z_t=S_t, as the authors point out. It might make sense to discuss this aspect in more detail, as well as possible ways to regularize Z_t to ensure it will be the smallest possible and not end up equal to S_t. Similar arguments are true for the other two objectives. The MI objectives are not complete, as in reality there have to be additional constraints on Z_t. Maybe this is something the authors would want to cover in the main text.

**Time Spent Reviewing:**

6

---

> ### Author Response · Authors · 2021-08-06
> **Importance of sufficiency; combining objectives; full state coverage assumption; use of DPI in Lemma 1**
>
> Thank you for your thoughtful review.
>
> *Response to concerns*:
>
> “Is sufficiency the right formalism?”: We believe the sufficiency of a state representation is an important characteristic, though by no means the only important factor. Utilizing representations that are not sufficient to solve a task can lead to poor results (as shown also in our experiments), and therefore we believe it is important for the community to understand which representation learning methods lead to sufficient representations. Of course, that doesn’t mean this is enough to get a good representation in every case - characteristics such as disentanglement and minimality may be desirable as well. Studying sufficiency is a long-established problem in the state aggregation literature; we view our work as a step in the direction of bringing this perspective to bear on modern deep RL techniques. Our contribution is largely theoretical in nature; the experiments demonstrate that insufficiency *can* have an impact on deep RL performance. Of course, many other factors, including exploration and the difficulty of MI estimation in high-dimensional spaces (e.g., Oord et al. 2018, Belghazi et al. 2018), will influence the performance of these methods on various benchmarks in practice. We will add a discussion of this limitation to the paper.
>
> “The forward objective is the clear winner?”: We do not aim to promote the use of one objective over another, but rather to introduce the ideas and tools to analyze MI-based objectives for their sufficiency. Note that we assume in the paper that the reward is a function of only the current state (see footnote 2 on page 2 of the appendix), so adding an additional constraint that the representation predict the reward is actually redundant with maximizing $J_{fwd}$. As we discuss in the paper in Section  7, $J_{fwd}$ may be a sub-par choice in some situations because it is not task-aware, and therefore retains all information that could be relevant for any possible task. We leave for future work the problem of characterizing the sets of MDPs for which $J_{state}$ and $J_{inv}$ are sufficient, though we discuss some simple examples in Section 7.
>
> “Novelty of ideas”: While the definitions of $\pi^*$ and $Q^*$ sufficiency for state representations have been proposed in prior works, our contribution is to apply this analysis to the representations learned by the popular practice of maximizing MI objectives. To our knowledge, the prior work does not analyze MI-based representation learning objectives. We view this as a first step towards a better theoretical understanding of these techniques which have performed well in practice (e.g., Anand et al. 2019, Stooke et al. 2020).
>
> *Response to questions*:
>
> “Combine various objectives?”: From a theoretical perspective, if all the joint objectives are maximized, the maximizing set is the intersection of the maximizing sets for each objective. It would be interesting future work to attempt to relate the amount of information in the representation learned by approximate MI-maximization to the performance of the resulting RL policy for representations that do not maximize the MI objective. However, we suspect this requires making additional assumptions on the MDP, since one can design an MDP in which missing a single bit of state information is catastrophic.
>
> “Why is full state coverage required in the dataset?”: We assume a fixed data collection policy in order to isolate the representation learning problem from that of exploration. The assumption that the policy has full coverage of the state space is a standard assumption in RL theory (Sutton & Barto 1998) and offline RL (Ernst et al. 2005). Without this assumption, we cannot say anything at all about the representation of states outside the support, and so it is not possible to guarantee sufficiency for any representation learning method.
>
> “Direction of Markov chain in Lemma 1”: The Data Processing Inequality relies on conditional independence between variables, also called D-separation. Since X -> Y -> Z and X <- Y -> Z have the same conditional independence relationships, DPI applies to both. We reproduce the proof from Chapter 2 of Cover & Thomas here: $I(X; Y, Z) = I(X; Z) + I(X; Y | Z) = I(X; Y) + I(X; Z | Y)$ by the chain rule for mutual information. By conditional independence, $I(X; Z | Y) = 0$. Since $I(X; Y | Z) \ge 0$, $I(X, Y) \ge I(X, Z)$.
>
> “Information bottleneck in addition to MI objective”: You are correct that each objective is maximized by the identity representation; however, other representations are also maximizers. In the deep RL experiments, we apply an information bottleneck of the form $I(S; Z)$ in order to learn the most compressed representation that still maximizes each objective. We will make this more clear in the text.

---

> > ### Author Response · Authors · 2021-08-16
> > **Any other concerns?**
> >
> > Dear reviewer,
> >
> > Please let us know if our responses have adequately addressed your concerns, and if you have further questions we can address. Thanks to your review, we will fix the typo regarding the Markov chain in Lemma 1 and justify the use of DPI via conditional independence, and clarify the text of the paper regarding the usefulness of sufficiency as a metric, the assumption of full state support in the dataset, and the potential downsides of $J_{fwd}$.

---

> > > ### Comment · Reviewer_Ncmr · 2021-08-17
> > > **Updated review**
> > >
> > > Thank you for detailed response. Yes, please add these clarifications above to the main text. Regarding Lemma 1, my recommendation is that you use the correct direction in the arrow to be consistent with the main text, but then explain why the claim holds by making use of the data processing inequality and the chain rule for mutual information.

---

### Official Review · Reviewer_w9XU · 2021-07-31

**Rating:** 7
**Confidence:** 3

**Summary:**

This paper discusses the sufficiency of several objectives for representation learning. It shows that learning representation by maximizing forward information is sufficient for RL under their definition and other two objectives are not. It also uses experiment to show that forward-information maximization is useful in RL in the sense that using it can train an agent that is at least as good as an end-to-end learner and can be better in some cases.

**Limitations And Societal Impact:**

The authors have addressed the limitations of their paper clearly.

**Main Review:**

Using sufficiency to analyze MI objectives is novel, and can enlighten future research on theory of representation learning and algorithm design. The paper is written clearly. Their theoretical results and experiments clearly show the value of using forward information for representation learning. Although I am not pretty sure whether I have missed something, to my knowledge, their proof, including the proof of Lemma 2, is now corrected.

**Time Spent Reviewing:**

5

---

> ### Author Response · Authors · 2021-08-06
> **Thank you for your review**
>
> Thank you for your positive comments! Please let us know if there are any questions you have that we can address.

---

### Decision · Program_Chairs · 2021-09-28

**Decision:**

Accept (Poster)

**Comment:**

All the reviewers found the paper worthy of acceptance at NeurIPS and I am happy to accept it. I encourage the authors to include in the final version the clarifications that they added in the rebuttal.

**Consistency Experiment:**

NeurIPS has a long history of experimentation. In 2014, NeurIPS ran an experiment in which 10% of submissions were reviewed by two independent committees to quantify the randomness in the review process. This year, we repeated a variant of this experiment to see how the quality of the review process has changed over time.  This paper was part of the experiment and was therefore assigned to two committees (consisting of reviewers, an Area Chair, and a Senior Area Chair) that reached independent decisions.  If both committees made the same recommendation, this recommendation was followed. If a single committee recommended acceptance, the paper was accepted (with the exception of a few cases in which the other committee identified what we considered a fatal flaw, e.g., an error in a key result).

This copy’s committee reached the following decision: **Accept (Poster)**

The other committee assigned to the paper recommended **Reject**.  You can find the other set of reviews, along with any follow up discussion with the authors here:
https://openreview.net/forum?id=haSQRA5RnuM